# Antipsychotic-induced extrapyramidal side effects: A systematic review and meta-analysis of observational studies

**Tilahun Ali**[1]*, **Mekonnen Sisay**[2], **Mandaras Tariku**[1], **Abraham Nigussie Mekuria**[2], **Assefa Desalew**[3]

**1** Department of Psychiatry, School of Nursing and Midwifery, College of Health and Medical Sciences, Haramaya University, Harar, Ethiopia, **2** Department of Pharmacology and Toxicology, School of Pharmacy, College of Health and Medical Sciences, Haramaya University, Harar, Ethiopia, **3** School of Nursing and Midwifery, College of Health and Medical Sciences, Haramaya University, Harar, Ethiopia

* tilahun1989@gmail.com

## Abstract

### Background

Antipsychotic agents are the basis for the pharmacological management of acute and chronic schizophrenia, bipolar disorders, mood disorders with psychotic feature, and other psychotic disorders. Antipsychotic medication use is frequently associated with unfavorable adverse effects such as extrapyramidal side effects (EPSEs). Hence, this systematic review and meta-analysis was aimed to determine the magnitude of antipsychotic-induced EPSEs.

### Method

A literature search was conducted using legitimate databases, indexing services, and directories including PubMed/MEDLINE (Ovid®), EMBASE (Ovid®), google scholar and World-Cat to retrieve studies. Following screening and eligibility, the relevant data were extracted from the included studies using an Excel sheet and exported to STATA 15.0 software for analyses. The Random effects pooling model was used to analyze outcome measures at a 95% confidence interval. Besides, publication bias analysis was conducted. The protocol has been registered on PROSPERO with ID: CRD42020175168.

### Result

In total, 15 original articles were included for the systematic review and meta-analysis. The pooled prevalence of antipsychotic-induced EPSEs among patient taking antipsychotic medications was 37% (95% CI: 18–55%, before sensitivity) and 31% (95% CI: 19–44%, after sensitivity). The prevalence of antipsychotic-induced parkinsonism, akathisia, and tardive dyskinesia was 20% (95% CI: 11–28%), 11% (95% CI: 6–17%), and 7% (95% CI: 4–9%), respectively. To confirm a small-study effect, Egger's regression test accompanied by funnel plot asymmetry demonstrated that there was a sort of publication bias in studies reporting akathisia and tardive dyskinesia.

**Data Availability Statement:** All relevant data are within the paper and its Supporting information files.

**Funding:** The author(s) received no specific funding for this work.

**Competing interests:** The authors have declared that no competing interests exist.

**Abbreviations:** EPSEs, Extrapyramidal side effects; JBI, Joanna Briggs Institute; PICOS, Participant, Interventions/Exposure/, Comparison, Outcome, and Study setting; SSA, Sub Saharan Africa.

## Conclusion

The prevalence of antipsychotic-induced EPSEs was considerably high. One in five and more than one in ten patients experienced parkinsonism and akathisia, respectively. Appropriate prevention and early management of these effects can enhance the net benefits of antipsychotics.

## Introduction

Antipsychotic agents are the foundation of the pharmacological management of acute and chronic schizophrenia, bipolar mood disorders, and other psychotic disorders [1]. They are classified as first-generation antipsychotics (FGAs), also called conventional (typical) agents and second-generation antipsychotics (SGAs) which also termed as atypical agents [2]. Like other medications, antipsychotic medications have both beneficial and harmful effects at the optimum dose used for the treatment purpose [3]. One of the major side effects of antipsychotics is extrapyramidal side effects (EPSEs), as they can cause distress and worsening psychopathological states [4, 5]. EPSEs also called drug-induced movement disorders include a wide variety of movement disorders and can be classified into acute and tardive symptoms, such as parkinsonism, dystonia, tardive dyskinesia (TD), and akathisia. These are a major class of neurologic adverse effects associated with antipsychotic medications with the typical agents taking the overwhelming majority [6, 7]. This side effects can stigmatize and cause patients' subjective distress, both of which are disincentives to continue to take medication. Moreover, they can confound the clinical assessment of negative symptoms of schizophrenia [8] and may lead to misdiagnosis and false conclusion that an increase in dose of antipsychotic medication is required, further exacerbating the problem.

An acute syndrome occur within days or weeks after starting an antipsychotic, or after increasing the dose and the tardive syndrome might be developed after months or years of antipsychotic treatment [9]. TD has been proposed to be caused by a relative cholinergic deficiency secondary to super-sensitivity of dopamine receptors in the striatum (caudate-putamen complex). Broadly, proposed mechanisms underlying these adverse events include decreased dopamine concentrations in the striatal area and dopamine hypersensitivity for drug-induced parkinsonism and TD, respectively [10, 11]. Pharmacologic treatment approaches for drug-induced parkinsonism have commonly included centrally acting anticholinergic agents such as benztropine; however, anticholinergic medications can make TD worse. Hence, switching the antipsychotic medication to those with lower propensity for parkinsonian and TD symptoms is worth mentioning [11].

Research-based evidence reported that the prevalence of antipsychotic-induced movement disorders among patients on long-term treatment with FGAs was around to be 50 to 75% [12]. SGAs such as quetiapine, olanzapine, and risperidone, have a lower liability for developing EPSEs compared with FGAs but there is some evidence of a lower risk of TD. This is because SGAs have looser interaction (lower affinity) with Dopamine (D2) receptor in the striatum [12]. Nevertheless, even with these newer agents, movement disorders are seen in a significant proportion of patients [13]. A meta-analysis of twelve trials, conducted in 2008, revealed that the annualized TD incidence was 3.9% for SGAs and 5.5% for FGAs. This data supports the relatively lower TD risk of SGAs. However, this study considered only one of the EPSEs (i.e. TD) and included clinical trials [14]. The safety of antipsychotic drugs in natural settings has been ignored. Another peer-reviewed medical literature showed that akathisia was reported to

occur in 25.9% of patients [15]. A study conducted with patient medical charts at a medical center in southern Taiwan showed that, of the 123 included subjects, 35 (28.5%) were found to have at least one episode of the tardive syndrome. The prevalence of subtypes of the tardive syndrome was 21.1%, 12.5%, 2.4% and 2.4% for tardive dystonia, tardive tremor, TD and tardive akathisia, respectively [16]. Generally, the use of atypical antipsychotics may have reduced EPSEs but findings are inconsistent and drug-specific as well as available evidence are quite disagreeing each other especially for TD and akathisia [17]. Besides, SGAs differ in many properties including safety, efficacy, cost and other pharmacological profiles. Hence it is too early to conclude about their EPSE profiles [18]. Meta-analysis of studies at natural treatment settings is also worth considering to explore the magnitude of the problem in real settings.

Hence, the main aim of this study is to provide conclusive and comprehensive evidence on the prevalence of antipsychotic medication-induced EPSEs and to formulate recommendations for policymakers, programmers, researchers, and clinical practice.

## Methods

### Study protocol and registration

The Preferred Reporting Items for Systematic Review and Meta-analysis (PRISMA) guideline was used for screening and eligibility evaluation of retrieved studies for systematic review and meta-analysis [19]. This systematic review and meta-analysis was conducted in accordance with the PRISMA Protocol [20]. In addition, the entire content of this systematic review and meta-analysis has been well reported in page by page in the completed PRISMA checklist (S1 Table). The study protocol was registered on the International Prospective Register of Systematic Reviews (PROSPERO) with a unique ID: CRD42020175168 and available online at: https://www.crd.york.ac.uk/PROSPERO/display_record.php?ID=CRD42020175168&ID= CRD42020175168.

### Data sources and search strategy

An electronic search was performed on legitimate databases and search engines including PubMed/ MEDLINE (Ovid), and EMBASE (Ovid), PsycInfo with predefined keywords, indexing and Medical Subject Headings (MeSH) terms until March 31st, 2020. The keywords and MeSH terms used for searching were "antipsychotic agents", "neuroleptic", "major tranquilizers", antipsychotic*, "extrapyramidal side effects", "extrapyramidal symptoms", EPS "movement disorders" [MeSH], EPSE, "motor disorders" [MeSH]. Searching on Google Scholar and WorldCat was also undertaken to identify other relevant works published in journals not indexed in legitimate databases and indexing interfaces, and to retrieve unpublished works (grey literature) including thesis/dissertations, institutional repositories, and organizational manuals, among others. Boolean operators (AND, OR) and truncation were used when appropriate to fine-tune the search and increase the chance of addressing relevant findings. Additionally, we back-traced the reference lists of retrieved articles to identify further relevant studies.

### Inclusion and exclusion criteria

During the screening and eligibility processes, there were predefined inclusion and exclusion criteria. Observational studies (cross-sectional or cohort) addressing antipsychotic-induced EPSEs in natural settings were included. Restrictions were applied on the year of publication (i.e., 2000 and onwards) and the language of publication (articles written in English) for the sake of considering methodological updates, time sensitivity and consistency of studies.

During the screening process of titles and abstracts, unrelated (off-target) articles, review papers, editorials, commentaries, opinions, qualitative studies, case reports and case series were excluded. Studies addressing drug induced-EPSEs in animal model (preclinical) as well as experimental (interventional) studies were also excluded during the selection process. We also excluded studies with incomplete information and those with mixed or non-specific outcome measures during the eligibility evaluation. At this stage, studies having irretrievable full texts (after requesting full texts from the corresponding authors via email and/or Research Gate account); studies with unrelated or insufficient outcome measures and studies with missed or ambiguous outcomes of interest were excluded.

## Screening and eligibility process

The retrieved citations from various databases and search engines were imported to END-NOTE version 7.2 software (Thomson Reuters, Stamford, CT, USA) with compatible file formats. Then, all citation lists from the aforementioned data sources were combined in one folder. Duplicate records were identified, recorded, and removed with the help of ENDNOTE software followed by careful visual inspection considering distinct referencing styles of sources in which the software could not detect as duplicate. Each record was independently assessed by two authors (TA and MS) using the predefined inclusion and exclusion criteria stated above. Following the initial screening of records with their titles and abstracts, a rigorous eligibility assessment of full texts was performed. Disagreements raised among authors during screening and eligibility process were solved with discussion and consultation of the rest authors (AD and ANM).

## Data extraction

Data extraction format prepared in Microsoft Excel sheet (S2 Table) was developed to extract data from each included study. The two authors (TA and MT) independently extracted data related to methodological characteristics and outcome measures including first author, publication year, study design and populations, study settings and country, overall drug regimen, sample size, type of EPSEs and/or their components.

## Critical appraisal of studies (risk of bias assessment)

Following the assessment of eligible articles, two authors (TA and ANM) independently assessed the methodological validity and analysis of outcome measures using the Joanna Briggs Institute (JBI) critical appraisal checklist for observational studies, University of Adelaide, Australia [21]. This critical appraisal was conducted to assess the internal (systematic error) and external (generalizability) validity of studies and to determine the extent to which each study has addressed the possibility of bias in its design, conduct, and analysis. The assessment tool consisted of ten and eleven design-specific questions for cross-sectional and cohort studies, respectively. They are designed to address the quality of the study for which articles were given one of the following responses for each question: Yes, No, Unclear or Not applicable. The mean score of the two authors was taken for the final decision and studies with a score of 'Yes' greater than or equal to half of the respective number of appraisal questions were included in the study.

## Outcome measurements

In this meta-analysis, the primary outcome of interest was the magnitude of antipsychotic medications-induced overall motor disorders/EPSEs and/or one of the components (i.e.,

parkinsonism, akathisia, and TD). Subgroup analyses were undertaken based on the geographic location and study setting.

## Data processing and analysis

The extracted data were exported from Excel to STATA 15.0 software (Stata Corporation, College Station, TX, USA) for analyses of overall outcome measures, subgroups and publication bias. Considering the variation in true effect sizes across the population, the inverse variance (IV) method with random effects pooling model was applied for meta- analysis at a 95% confidence level. The heterogeneity of studies was assessed using $I^2$ statistics. The "leave-one-out" sensitivity analysis was carried out to assess outliers that likely have a substantial impact on the overall effect size and between study heterogeneity [22]. The presence of publication bias was determined by visualization of funnel plot asymmetry supplemented with Egger's regression test [23, 24]. Statistical tests were declared significant at cutoff point of $p < 0.05$ (two-sided).

## Results

### Search findings

A total of 1933 studies were retrieved through visiting legitimate electronic databases, indexing services, search engines, and repositories. From these, 934 duplicate studies were identified and removed using ENDNOTE followed by careful visual inspection. Then, 999 records were retained for further screening of their titles and abstracts. Among which, a total of 941 records (723 studies with titles and 218 studies with abstracts) were excluded. The full texts of the remaining 58 studies were thoroughly assessed for their eligibility and 43 of which were excluded for various reasons such as studies that didn't report the study outcome, having incomplete information, and/or those reported vague or mixed findings. Finally, 15 studies were included in the systematic review and meta-analysis (Fig 1).

### Results of critical appraisal

Studies that fulfilled the inclusion criteria were subjected for further rigorous appraisal using the JBI checklist for observational studies (cross-sectional and cohort studies with 9 to 11-scaled questions, respectively). During the quality assessment, the average quality scores of individual studies ranged between 6 and 10. Finally, 15 studies were included in systematic review and meta-analysis (S3 Table).

### Study characteristics

A total of 15 studies were included in this systematic review and meta-analysis. The publication years of the included studies ranged from 2000 to 2019. Regarding the geographical distribution, the review included six studies from Europe [4, 25–29], five studies from Asia [30–34], two studies from Africa [35, 36], one study from South America [37], and one study from North America [38]. Thirteen of these studies had employed cross-sectional design [4, 26, 28–31, 33–37] with two of which being a retrospective chart review [25, 32] and the remaining two studies had employed cohort design [27, 38]. The sample size of individual studies ranged from 28 [29] to 164,417 [38]. Thirteen studies were conducted in hospital settings [4, 26–31, 33–38] whereas two studies were conducted at rehabilitation centers [25, 32]. Majority of studies were conducted on adult patients with schizophrenic or schizoaffective disorders whereas the rest studies were conducted on non-specific psychotic patients, acute psychotic patients with substance use disorder, and patients with severe mental illness. Regarding the regimen, seven studies reported mixed use of both typical and atypical antipsychotics [25, 27–30, 33, 38], five

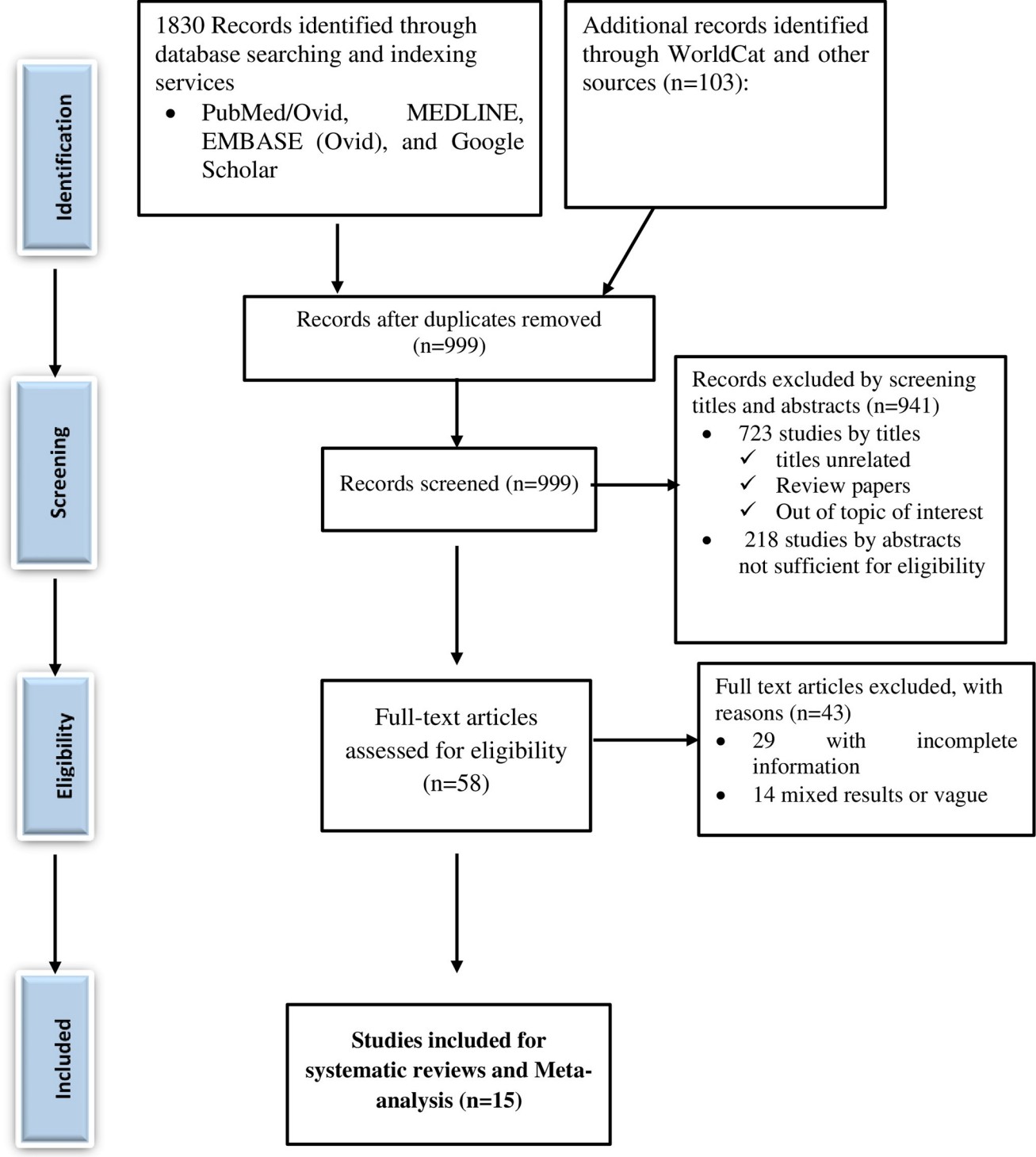

**Fig 1. PRISMA flow diagram depicting the selection process of identified studies.**

studies reported independent use of either oral or depot (IM) typical antipsychotics (primarily phenothiazines and thioxanthenes) [4, 31, 32, 35, 36] and three studies included participants who took atypical agents only [26, 34, 37]. Regarding the outcome measures, twelve studies reported the overall magnitude of antipsychotic-induced EPSEs [25, 26, 28–37]. Looking at

individual components of EPSEs, nine, nine and eight studies reported drug-induced parkinsonism, akathisia and TD, respectively. All included studies utilized standardized measurement scales such as Barnes Akathisia Scale [39], Modified Simpson-Angus Scale [40] or Unified Parkinson Disease Rating Scale (UPDRS) [41], and Abnormal Involuntary Movement Scale (AIMS) [42] for rating akathisia, parkinsonism, and TD, respectively (Table 1).

## Meta-analysis of outcome measures and sensitivity analysis

For estimating the overall magnitude of antipsychotic-induced EPSEs, 12 studies reporting the outcome measures were included for meta-analysis. Based on this, the pooled estimate was found to be 37% (95% CI: 18–55%). Following the 'leave-one-out' sensitivity analysis, the pooled EPSE estimate was considerably reduced to 31% (95% CI: 19–44%) with little change (1.00%) in the degree of heterogeneity (Fig 2A and 2B).

## Subgroup analysis

Analysis stratified based on settings indicated that those patients treated at the hospital showed higher pooled antipsychotic-induced EPSEs, 33% (95% CI: 17–49%) than those treated at rehabilitation center, 13% (95% CI: 9.0–17%). Besides, subgroup analysis based on continent indicated that patients from Africa showed relatively higher drug-induced EPSEs, 51% (47–56%) whereas the prevalence was by far lower from Asia, 13% (95% CI: 4.0–21%) (Fig 3A and 3B).

## Meta-analysis of EPSE components

Nine studies reported antipsychotic-induced parkinsonian symptoms. The pooled estimate of these studies was found to be 20% (95% CI: 11–28%) (Fig 4). Likewise, nine studies either alone or with other EPSE components reported the prevalence of antipsychotic-induced akathisia. Based on this, the meta-analysis of these studies showed the pooled estimate of akathisia was11% (95% CI: 6.0–17%) (Fig 5). What is more, eight studies reported tardive dyskinesia as one of the EPSEs associated with antipsychotic medications. In this regard, the pooled prevalence of tardive dyskinesia was about 7% (95% CI: 4–9%) (Fig 6).

## Publication bias

To confirm a small-study effect, Egger's regression test accompanied with funnel plot asymmetry demonstrated that there was a sort of publication bias in studies reporting akathisia (Egger's Q = 6.43, P = 0.019) and tardive dyskinesia (Egger's Q = 3.88, P = 0.021) (Fig 7A–7D).

# Discussion

This systematic review and meta-analysis determined the pooled proportion of antipsychotic medications-induced EPSEs among patients in natural treatment settings. The pooled prevalence of antipsychotic-induced EPSEs among patient taking these medications was 37% and 31% in before and after sensitivity analysis, respectively. The pooled prevalence of antipsychotic-induced parkinsonism, akathisia, and TD were 20%, 11%, and 7%, respectively.

Anti-psychotic medications have proven efficacy in the treatment of schizophrenia and other major psychiatric conditions. Although the clear mechanism of action remains elusive and gradually evolving, most antipsychotics have shared their action on dopaminergic and serotoninergic neurotransmission. The effectiveness of antipsychotic medications depends on the activity and the specific location of these neuronal receptors. The potency of conventional (typical) antipsychotic drugs correlates closely with their affinity for the dopamine 2 (D2) receptor, blocking the effect of endogenous dopamine in different dopaminergic pathways [43].

**Table 1. Summary of studies included in the systematic review and meta-analysis.**

| Author, year | Country | Design | Participants | Setting | Drug and regimen | Sample size | Overall EPSEs | TD | Akathisia | Parkinsonism |
|---|---|---|---|---|---|---|---|---|---|---|
| Araújo A. et al., 2016 [37] | Brazil | CS | Schizophrenic patients (age:18–65 years) | Hospital | Atypical antipsychotics (Olanzapine, risperidone, ziprasidone, Clozapine) | 213 | 81 | NS | NS | NS |
| Dhavale H. et al., 2004 [31] | India | CS | Psychiatric patients | Hospital | Haloperidol 2-13mg/day equivalent for 02 months | 71 | 68 | NS | NS | NS |
| Duangrithi D. et al., 2016 [32] | Thailand | CS (R) | Acute psychotic patients with SUD | Rehabilitation center | Depot (IM) antipsychotic agents (Haloperidol or Zuclopenthixol or combination) | 153 | 12 | NS | 7/153 | NS |
| Gebhardt S.et al., 2006 [25] | Germany | CS (R) | Adolescent psychotic patients | Rehabilitation center | Atypical (81.7%) or typical (10.8%) 0r combination (7.5%) for greater than one year | 93 | 37 | 5 | 1 | 2 |
| Ghoreishizadeh M. and Deldoost F. 2008 [34] | Iran | CS | Schizophrenic and Schizoaffective patients (age: 18–60 years) | Hospital | Risperidone 2mg/day for 6 weeks | 100 | 28 | | 5 | 23 |
| Luft B. and Berent E., 2009 [26] | UK | CS | Psychotic patients | Hospital | Long acting (depot) antipsychotics (Risperidone) inj. | 43 | 23 | 10 | 12 | 13 |
| Mentzel L., et al., 2017 [33] | Netherlands | CS | patients with severe mental illness | Hospital | Both typical and atypical (not specified) | 191 | 35 | NS | NS | NS |
| Moreno-Calvete MC. 2013 [29] | Spain | CS | Schizophrenic patients (> 18 years) | Hospital | Both typical and atypical (not specified) for greater than 6 months | 28 | 6 | 1 | 2 | 4 |
| Ojagbemi A.et al., 2018 [36] | South Africa & Nigeria | CS | Schizophrenic patients (> 18 years) | Hospital | Flupenthixol decanoate (< = 30 mg) for 03 months | 99 | 34 | NS | NS | NS |
| Taye H.et al., 2014 [35] | Ethiopia | CS | Psychotic outpatients | Hospital | Chlorpromazine (> = 400 mg/day) equivalent | 377 | 212 | 45 | 108 | 175 |
| Desai N. et al.2017 [30] | India | CS | Psychiatric patients (20–60 years) | Hospital | Monotherapy primarily atypical and the rest typical and combination therapy | 706 | 40 | 4 | 6 | 36 |
| Loughlin AM. et al., 2019 [38] | USA | Cohort study (R) | All antipsychotic users (> 18 years) | Database | Antipsychotic agents not specified | 164,417 | NS | 1,314 | NS | NS |
| Misdrahi D. et al., 2019 [27] | France | Cohort | Schizophrenic patients (> 16 years) | Hospital | Monotherapy primarily atypical and the rest typical and combination therapy | 674 | NS | **56** | NS | 89 |
| Modestin J. et al., 2000 [28] | Switzerland | CS | All inpatients | Hospital inpatients | Typical (63.5%) or clozapine (23%) or both (13.5%) | 200 | 84 | 44 | 22 | 40 |
| Berardi D. et al., 2000 [4] | Italy | CS | Schizophrenic patients (20–69 years) | Community mental Health OPD | Chlorpromazine (430 ±337 mg) equivalent | 69 | NS | NS | 19 | 19 |

CS, cross-sectional; CS(R), cross-sectional with retrospective approach; SUD, substance use disorder; NS, not specified; EPSEs, extrapyramidal side effects; TD, tardive dyskinesia

**A)**

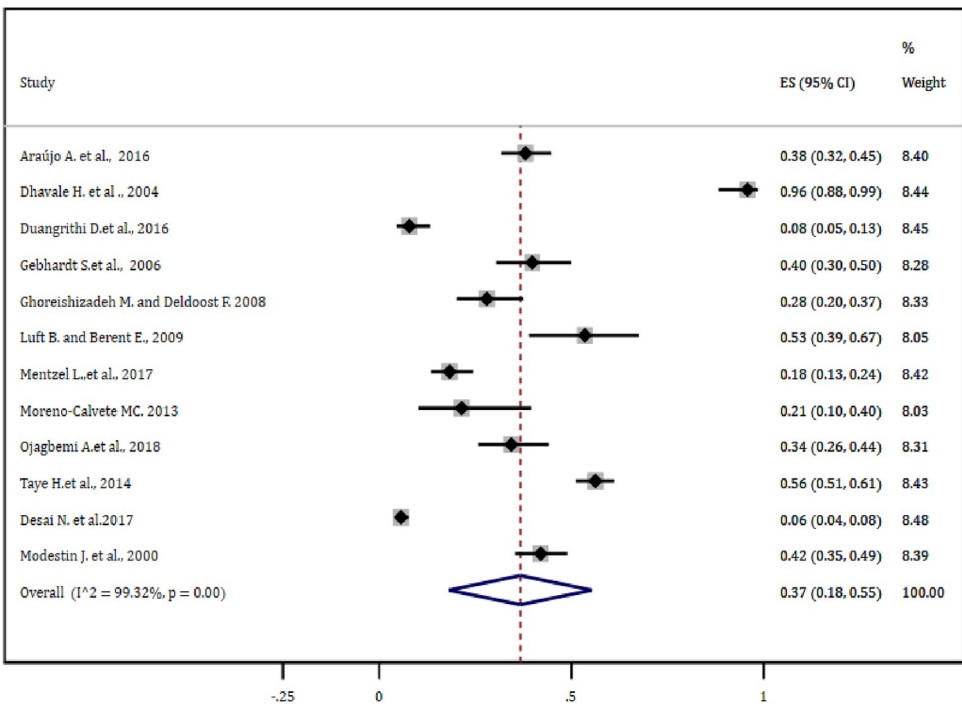

**B)**

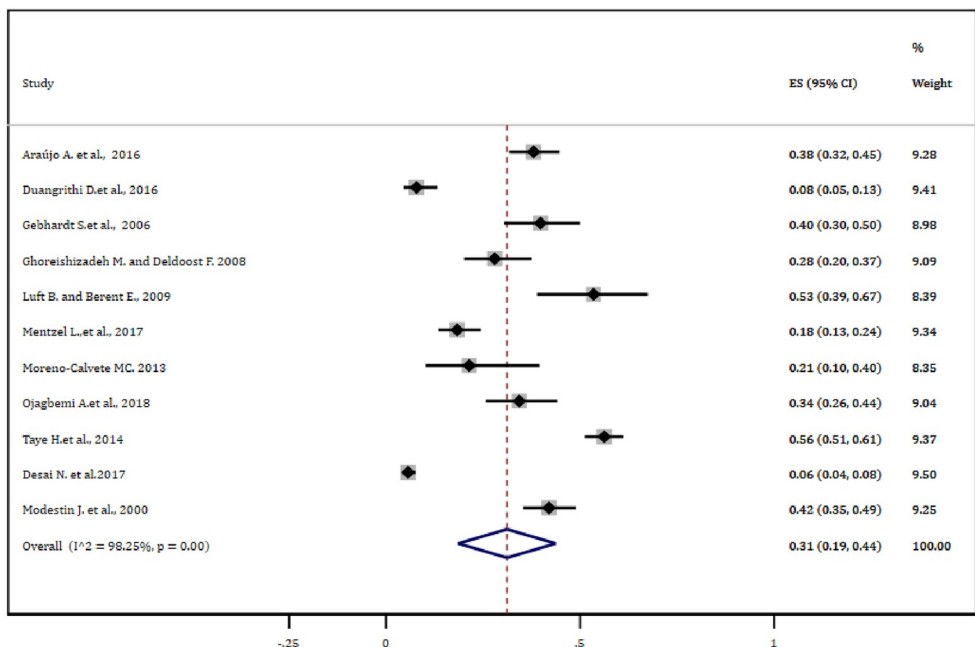

**Fig 2. Forest plot depicting the overall antipsychotic-induced EPSE (A) before and (B) after sensitivity analysis.**

Dopaminergic neurons have four major pathways in the CNS: mesolimbic (from the ventral tegmental area (mid-brain) to limbic system), mesocortical (mid-brain to the prefrontal cortex), nigrostriatal (substantia nigra to striatum), and tuberoinfundibular pathways (hypothalamus to anterior pituitary) [44]. Antipsychotic medication, particularly the conventional antipsychotics

A)

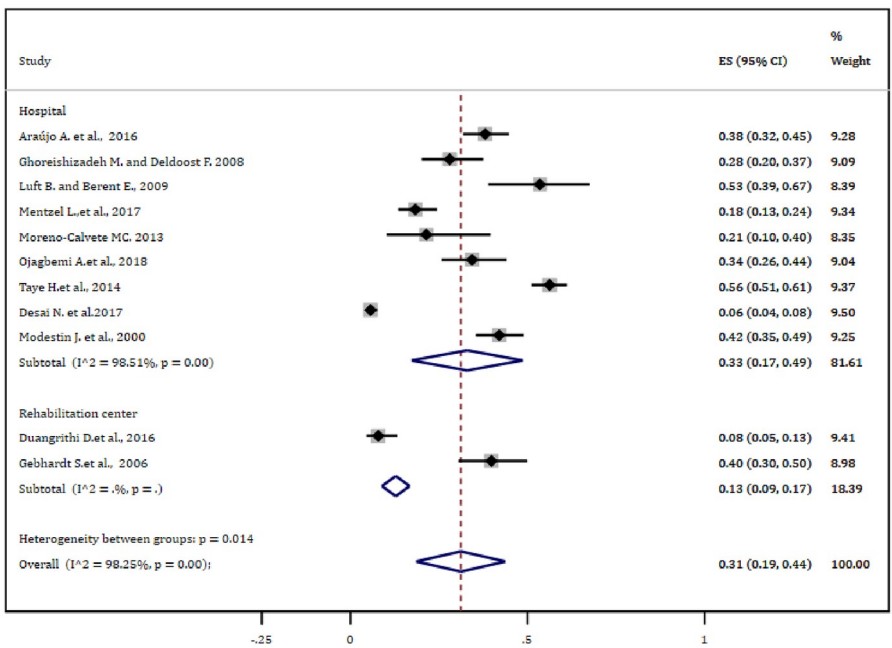

B)

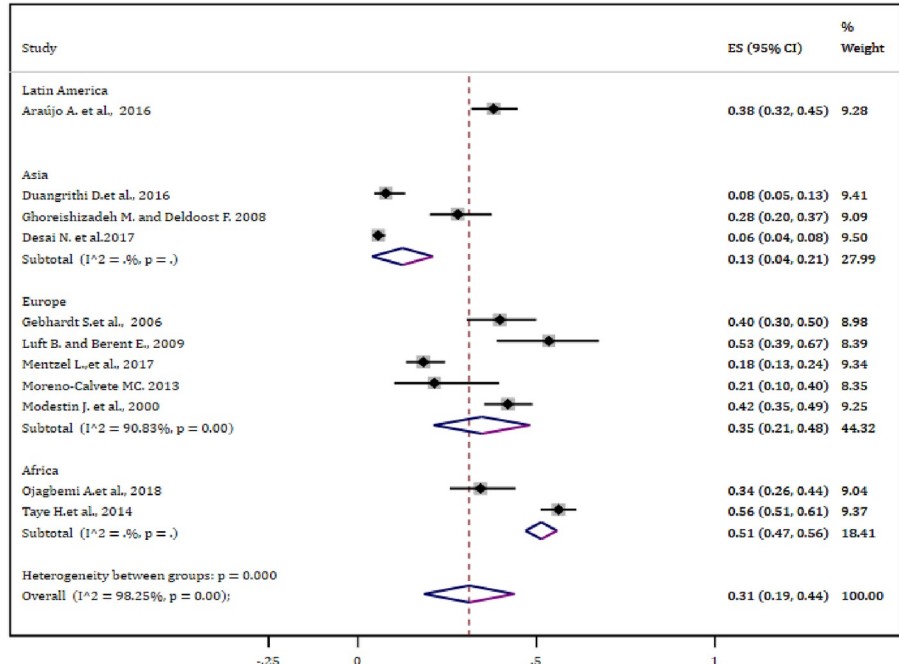

**Fig 3. Subgroup analysis of the overall EPSE estimate (A) Based on settings (B) Based on geographical distribution.**

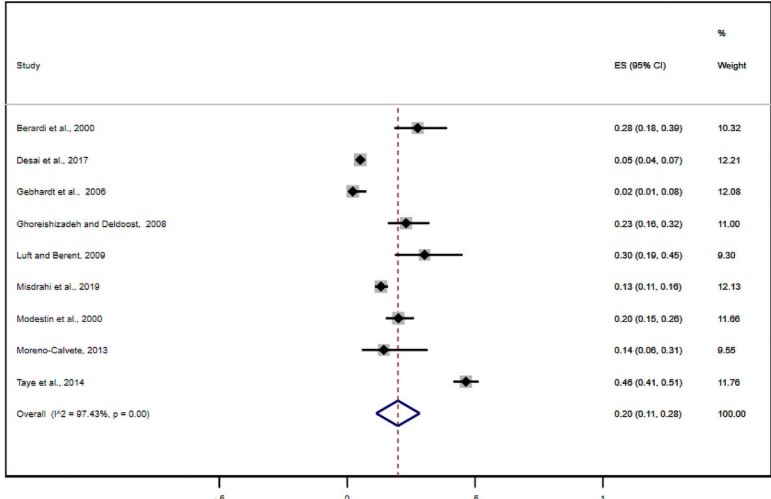

**Fig 4. Forest plot depicting anti-psychotic induced parkinsonism.**

have higher affinity for binding with dopamine receptors in all pathways. Blockade of dopamine receptors at mesolimbic and mesocortical pathways are responsible for the treatments of positive symptoms (hallucination, delusion, disorganized speech) and worsening of negative symptoms (avolution, alogia, anhedonia), respectively [45]. However, within the basal ganglia, the blockade of dopamine receptors at the striatum (caudate-putamen complex) (a receptor for dopaminergic neuron projecting from substantia nigra pars compacta (SNpc) to striatum) is primarily responsible for antipsychotic-induced EPSEs [46, 47]. This area is predominately involved in the coordination of voluntary movements and fine-tunes motor coordination between the thalamus and cortex. Hence, blockade of primarily D2 receptors (tonic inhibitor by nature) at striatum results in excitation of thalamocortical axis and brain stem motor nuclei.

Previous studies have revealed that the prevalence, incidence, and course of EPSEs associated with antipsychotic medications are varying, due to variability in the criteria to define and

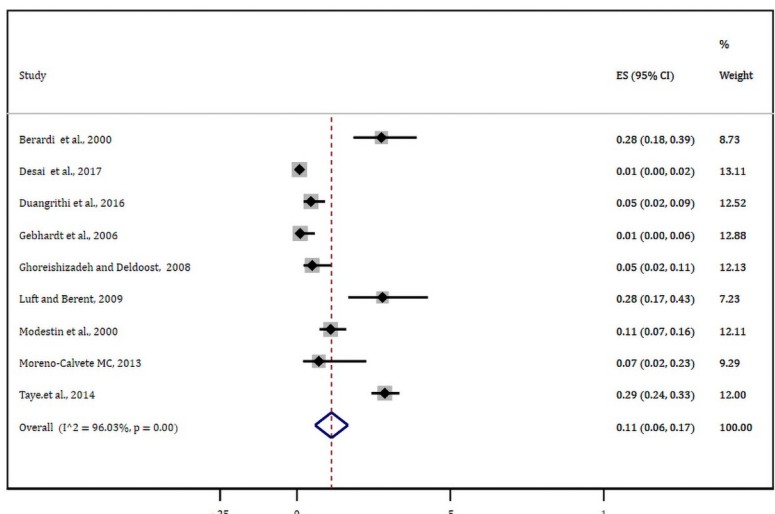

**Fig 5. Forest plot depicting anti-psychotic induced akathisia.**

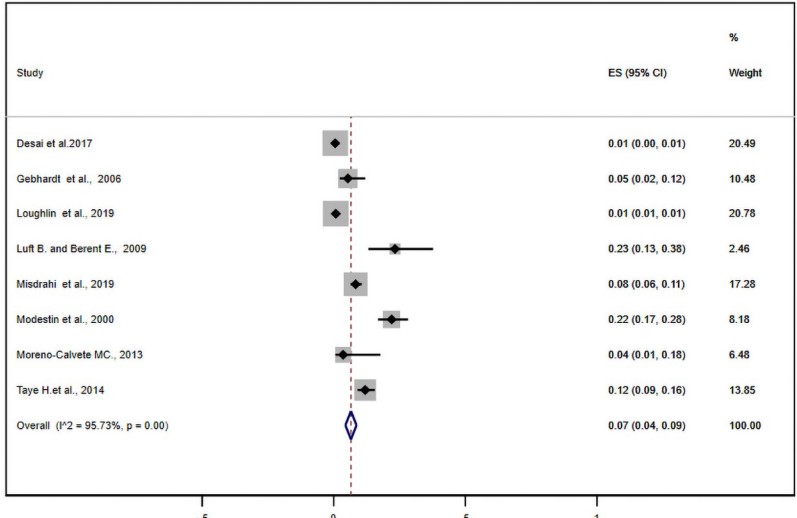

**Fig 6. Forest plot depicting anti-psychotic induced tardive dyskinesia.**

identify EPSEs among clinicians [48]. This finding was slightly higher compared to a population-based study conducted in Italy that estimated the prevalence of all common categories of movement disorders was 28% [49]. This might be due to the previous study was a single population-based study.

The finding from this study on TD is consistent with a systematic review which revealed the pooled prevalence of TD was 9.4% among patients taking antipsychotics medications [14]. The pooled estimate on akathisia was found lower as compared with a reviewed medical literature which stated that 25.9% of patients had developed akathisia [15]. The difference could be due to the type of mediations taken, meaning that the previous review included those patients taking all psychotropic medication, but this finding was restricted to those who took antipsychotic medications. The previous reviews also could not quantitatively analyze (meta-analyze) studies in natural treatment settings to generate evidence-based pooled estimate.

This study revealed that the pooled prevalence of TD was found to be 7% which is higher than a previously conducted TD-specific systematic review of 1-year studies on SGAs estimated that the weighted mean annual incidence risk of TD associated with these antipsychotics was 2.1% [50]. Such a difference might be ascribed to atypical antipsychotic drugs that may induce less dopamine hypersensitivity in the caudate-putamen complex. The prevalence of antipsychotic-induced parkinsonism is consistent with a previous literature review that summarized drug-induced parkinsonism and TD range from approximately 20 to 35% among antipsychotic users [11]. The previous study was a narrative review of antipsychotic-induced movement disorders with special focus on drug-induced parkinsonism and TD. Our study provided a pooled estimate of these movement disorders from systematically reviewed articles. Generally, to reconcile the paradox of tardive and parkinsonian symptoms, considering newer treatment options with lower neurological side effects is a game changer. Antipsychotic-induced TD is difficult to treat, frequently irreversible and with low tendency of remission. Drugs that target vesicular monoamine transporter -2 (VMAT 2) inhibitors have been found effective in the management of TD without worsening other EPSEs [51].

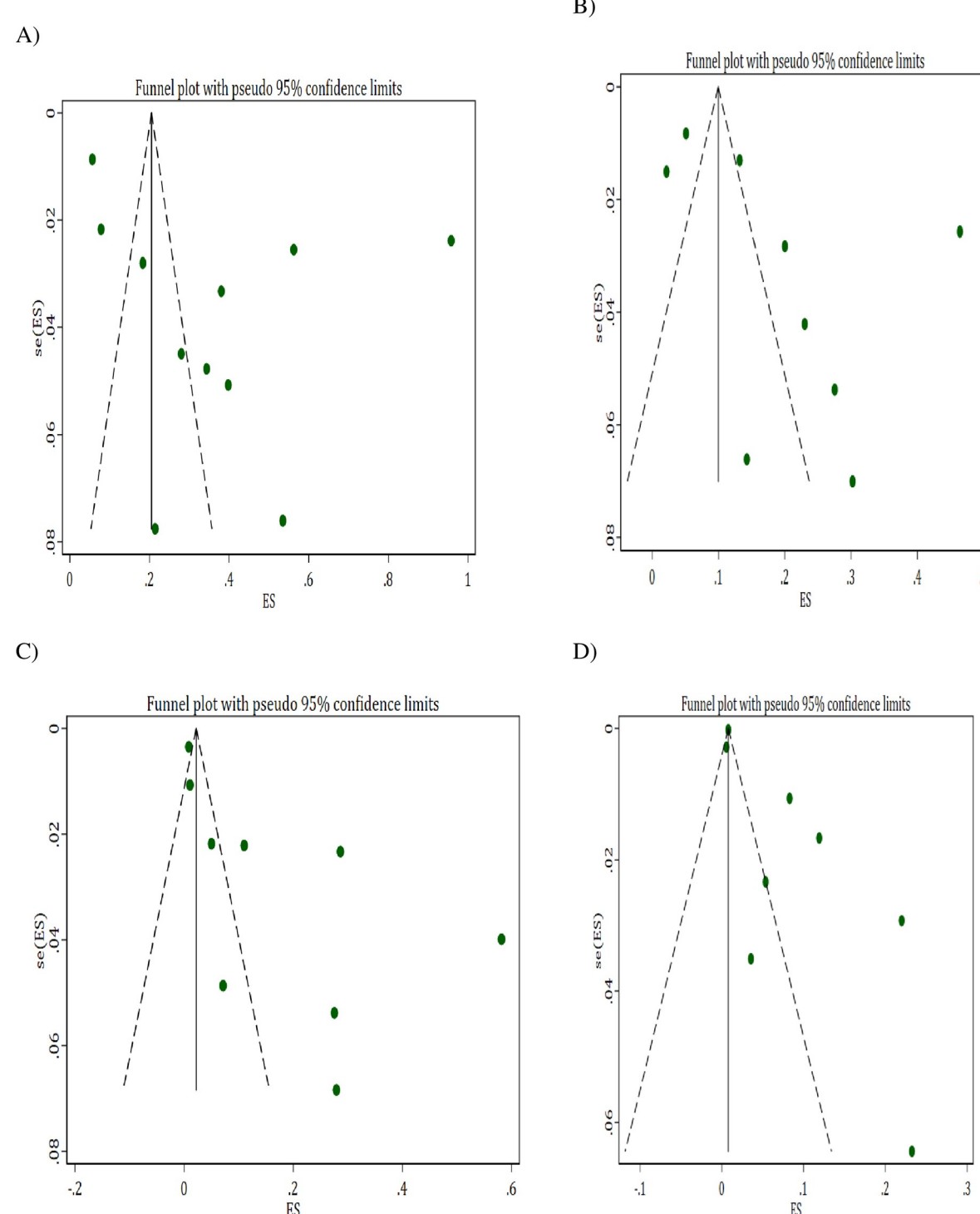

**Fig 7. Funnel plot depicting publication bias (A) Overall EPSEs (B) Parkinsonism (C) Akathisia and (D) Tardive Dyskinesia.**

## Strength and limitations of the study

### Strength

The previous studies considered only one of the EPSEs (i.e. TD, akathisia) and/or included mere clinical trials. The rest studies were narrative reviews that summarized various studies dealing about akathisia. However, such studies were not systematically selected. In this regard, the safety of antipsychotic drugs in natural settings would have been addressed. Meta-analysis of studies at natural treatment settings is also worth considering to explore the magnitude of the problem in real settings. What is more, the previous reviews also could not quantitatively analyze (meta-analyze) studies in natural treatment settings to generate evidence-based pooled estimate. This study primarily focuses on observational studies (neither preclinical nor interventional) as the experimental nature of trials in ideal settings may confound the pooled estimates of observational studies. Apart from this, most of the measurement scales for motor disorders came into clinical practice in late 1980s and mid1990s. Advances in pharmaceutical formulations in the area of antipsychotics should also be considered.

### Limitation

We have initially considered subgroup analysis on both demographic and clinical factors. Nevertheless, the high heterogeneity of diagnosis (Schizophrenic, acute psychotic, SMI, psychotic with SUD, etc.. .) and medication use (typical, atypical and mixed use) limited us from doing further meta-analysis. Instead, we have vividly included the age of the patients and drug regimen issues in the summary of studies. In the future, focused and drug-specific meta-analyses and scoping reviews as well as studies that address diverse methodological approaches should be considered for comparison.

## Conclusion

The prevalence of antipsychotic-induced EPSEs was considerably high. In this meta-analysis, the drug-induced parkinsonism was the most prevalent EPSE followed by akathisia. Besides, one in thirteen patients experienced TD which is a difficult to treat EPSE requiring special emphasis by clinicians. Treatment should consider the paradox of tardive and parkinsonian symptoms to the minimum. Appropriate prevention and early management of these side effects can enhance the net benefits of antipsychotic therapy. Designing EPSE treatment guideline, choosing those antipsychotics which had minimal side effects and psycho-education is worth considering.

## Supporting information

**S1 Table. Completed PRISMA checklist.** The checklist highlights the important components addressed while conducting systematic review and meta-analysis from observational studies. (DOC)

**S2 Table. Data abstraction format with crude data.** The table presented the ways of data collection (study characteristics and outcome measures) in Microsoft excel format. (XLSX)

**S3 Table. Critical appraisal scores of included studies.** The table shows the risk of bias assessments of studies with regard to design, conduct and analysis. (DOCX)

## Acknowledgments

We would like to address our deepest gratitude to the authors of the included studies for this systematic review and meta-analysis. Also, our deepest heartfelt goes to the staff of Haramaya University, College of Health and Medical Sciences who gave us technical support.

## Author Contributions

**Conceptualization:** Tilahun Ali, Mekonnen Sisay.

**Formal analysis:** Mekonnen Sisay.

**Methodology:** Tilahun Ali.

**Writing – original draft:** Tilahun Ali.

**Writing – review & editing:** Tilahun Ali, Mekonnen Sisay, Mandaras Tariku, Abraham Nigussie Mekuria, Assefa Desalew.

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
