## [Decision Letter · Decision Letter 0]

7 Dec 2020

PONE-D-20-32247

Antipsychotic Induced Extrapyramidal Side Effects : A systematic review and Meta-analysis

PLOS ONE

Dear Dr. Ali,

Thank you for submitting your manuscript to PLOS ONE. After careful consideration, we feel that it has merit but does not fully meet PLOS ONE’s publication criteria as it currently stands. Therefore, we invite you to submit a revised version of the manuscript that addresses the points raised during the review process.

We look forward to receiving your revised manuscript.

Kind regards,

Ahmed Negida, MD

Academic Editor

PLOS ONE

Journal Requirements:

2. Thank you for submitting the above manuscript to PLOS ONE. During our internal evaluation of the manuscript, we found significant text overlap between your submission and the following previously published works:

https://api.ithenticate.com/en_us/dv/0425?o=63584149&lang=en_us

https://bmcpharmacoltoxicol.biomedcentral.com/track/pdf/10.1186/s40360-019-0315-9.pdf

Please revise the manuscript to rephrase the duplicated text, cite your sources, and provide details as to how the current manuscript advances on previous work. Please note that further consideration is dependent on the submission of a manuscript that addresses these concerns about the overlap in text with published work.

Reviewers' comments:

Reviewer's Responses to Questions

**Comments to the Author**

1. Is the manuscript technically sound, and do the data support the conclusions?

Reviewer #1: Partly

Reviewer #2: Yes

Reviewer #3: Yes

2. Has the statistical analysis been performed appropriately and rigorously? 

Reviewer #1: No

Reviewer #2: No

Reviewer #3: Yes

3. Have the authors made all data underlying the findings in their manuscript fully available?

Reviewer #1: Yes

Reviewer #2: Yes

Reviewer #3: Yes

4. Is the manuscript presented in an intelligible fashion and written in standard English?

Reviewer #1: Yes

Reviewer #2: No

Reviewer #3: Yes

5. Review Comments to the Author

Reviewer #1: The authors present a systematic review and meta-analysis of antipsychotic-induced EPS. A few points to consider:

-The fourth paragraph (or second from end of section) of the introduction h isn't as informative compared to the large, meta-analyses in the preceding sentence because it is hard to tell why you are reporting these specific sentences.

-The rationale isn't entirely clear for the need for this meta-analysis when you cite large systematic reviews and patient studies on EPS. Consider being more clear for the need of your systematic review and meta-analysis

Methods

-are RCTs not an option for this type of analysis? Can you clarify your inclusion criteria in this regard?

-Why the restriction to the year 2000 and forward,

-so are you only including studies that had a primary outcome of EPS?

-I think it is rare to see quality cutoffs for inclusion into a meta-analysis. Can you cite where this methodology has been validated? Why would it be even useful to search the grey literature when it would likely fall below the cutoff with a lack of information?

-Did you consider sub-analyses by medication type? Diagnosis? Age?

-a table of characteristics of your final included studies is necessary.

-why 11 studies in the main meta-analysis when 15 included? More clarity needed.

-where are the I2 heterogeneity statistics throughout the results?

-more consideration and detail is needed regarding the included study antipsychotic type given their importance as detailed in the introduction.

Reviewer #2: Comments:

Title

1. Does the title give clear idea about the article? Yes

Abstract

2. Does the abstract concisely describe the content and scope of the project and identifies the project’s objective, its methodology and its findings, conclusions, or intended results? No

Under background, add something about mood disorder with psychotic. Because they also take antipsychotic medicine.

Under conclusion, you tried to conclude as appropriate prevention of EPSE is important. How you could come across with this recommendation?

Introduction

3. Does the introduction give clear idea about the article? YES

Please clarify the statement “They are classified as first generation or conventional (typical) antipsychotics (FGA) and second generation (atypical) antipsychotics (SGA)”

Use similar word for antipsychotic/s through the document. Also for EPS/EPSE. Use similar word and abbreviation. Sometimes you say EPS, and sometimes EPSE/s

What do you mean EPSs? The end of paragraph of introduction

What do you say by side effects which appear when anticholinergic medicine removed from the patient?

Methodology

4. Did methodology part is clear? No

Well, if specified who (author/s) were extracted data.

What do you say about data quality assessment?

Which model did you use to determine pooled prevalence?

Is all original paper report standard error? If no, what is your action? If yes, please specify.

What is about independent factors?

Results

5. Are results clear and appropriate with title? Yes

6. Revise the references as per the journal guideline

7. Generally, the paper is interesting. The most problems were found around methodology part. It needs major revision.

8. The paper needs an English language copy editing from the beginning to the end. Please focus on it.

Reviewer #3: I greatly appreciate the reviewing process and would like to propose designing a further review on the antipsychotic-induced dystonia and the relative contribution of each antipsychotic to the side effect specified in addition to the dose-related side effect threshold.

6. PLOS authors have the option to publish the peer review history of their article (what does this mean?). If published, this will include your full peer review and any attached files.

Reviewer #1: No

Reviewer #2: No

Reviewer #3: **Yes: **MM

---

## [Author Response · Author response to Decision Letter 0]

22 Jan 2021

Response to Academic Editor and reviewers 

Dear,

We would like to thank for your constructive comments to improve the manuscript titled “Antipsychotic Induced Extrapyramidal Side Effects: A systematic review and Meta-analysis” (ID: PONE-D-20-32247). Hereunder are our reaction to the academic editor and reviewers’ comments. 

Comments and questions Authors response

Academic Editor 

A rebuttal letter that responds to each point raised by the academic editor and reviewer(s). You should upload this letter as a separate file labeled ‘Response to Reviewers’.

A marked-up copy of your manuscript that highlights changes made to the original version. You should upload this as a separate file labeled ‘Revised Manuscript with Track Changes’.

An unmarked version of your revised paper without tracked changes. You should upload this as a separate file labeled ‘Manuscript’.

 Please ensure that your manuscript meets PLOS ONE’s style requirements, including those for file naming. 

 During our internal evaluation of the manuscript, we found significant text overlap between your submission and the following previously published works:

 Please include a separate caption for each figure in your manuscript.

 Please include captions for your Supporting Information files at the end of your manuscript, and update any in-text citations to match accordingly. 

An author response letter addressing comments forwarded from the academic editor and reviewer(s); revised manuscript with track changes that highlights changes made to the original version; and a clean version of the revised manuscript are separately presented as per the request. 

We prepared the revised manuscript as per the PLOS ONE’s formatting guidelines 

Textual overlap observed between the current manuscript and previously published works has been resolved with rephrasing such statements. 

A caption/titles for figures have been included in the manuscript as per the PLOS ONE requirement 

Captions for the supporting information files have also been included at the end of the manuscript (after the reference with specified labeling) 

Overall, the comments and suggestions provided by the academic editor is accepted and corrected accordingly.

We also incorporated nine relevant citation in the background, method and discussion sections to maintain the scientific integrity. Kindly check the track changed manuscript, please. 

We have also undergone careful language (mainly punctuation, syntax and normalization issues) and technical editions throughout the manuscript to make it clearer and easier to comprehend

We have strictly followed PLOS ONE formatting guidelines including 

o Correcting all figures using PACE (e.g. tif compatible format) with better resolution

o Rearranging the position of tables next to the paragraphs that they are first cited

o Formatting the title page, heading and subheadings of main text as per the guideline 

Reviewer 1

The authors present a systematic review and meta-analysis of antipsychotic-induced EPS. A few points to consider:

1. The fourth paragraph (or second from end of section) of the introduction isn't as informative compared to the large, meta-analyses in the preceding sentence because it is hard to tell why you are reporting these specific sentences.

2. The rationale isn't entirely clear for the need for this meta-analysis when you cite large systematic reviews and patient studies on EPS. Consider being more clear for the need of your systematic review and meta-analysis

 Dear, 

The points raised in the introduction part of the manuscript is revised deeply.

The previous study considered only one of the EPSEs (i.e. TD) and included clinical trials only.

The safety of antipsychotic drugs in natural settings has been ignored. Meta-analysis of studies at natural treatment settings is also worth considering to explore the magnitude of the problem in real settings. 

The other study was a narrative review that summarized various studies dealing about akathisia. However, such studies were not systematically selected. 

The previous reviews also could not quantitatively analyze (meta-analyze) studies in natural treatment settings to generate evidence-based pooled estimate. 

We have explicitly presented the rationale of this research in the background section in more elaborated manner. 

Methods

3. are RCTs not an option for this type of analysis? Can you clarify your inclusion criteria in this regard? 

Before 2010, meta-analysis of clinical trials revolving around the components of EPSEs (TD and akathisia) were conducted. Since then, no comprehensive EPSE based studies was conducted. 

This study primarily focuses on observational studies (neither preclinical nor interventional). The experimental nature of trials in ideal settings may confound the pooled estimates of observational studies. 

In this revised manuscript, we have tried to present quite elaborated rationale, inclusion criteria and discussion for what we conducted this meta-analysis. 

4. Why the restriction to the year 2000 and forward, Dear reviewer, 

We have considered methodological updates, time sensitivity and consistency of the research contents 

For instance, most of the measurement scales including Akathisia Rating Scale, Unified Parkinson Disease Rating Scale, Abnormal Involuntary Movement Scale (for TD) came in clinical practice in late 1980s and mid1990s. 

In the preliminary screening process, majority of studies about this research topic were conducted in the last 20 years (21st century).

Advances in pharmaceutical formulations in the area of antipsychotics should also be considered 

In our case, we finally included 60% studies from 2010-2019 and 40% from 2000-2009 after PRISMA flow chart and quality assessments. 

5. so are you only including studies that had a primary outcome of EPS?

-I think it is rare to see quality cutoffs for inclusion into a meta-analysis. Dear, 

As clearly explained in the methodology section, original studies addressing the magnitude of EPSE and/or one of its components (e.g., TD, DIP and/or akathisia) were included in the study once they passed the quality assessment.

It doesn’t mean that having any of the EPS component give rise the entire EPS since one patient may experience one or all of the symptoms at a time

The authors extracted the data meticulously with several check points 

6. Can you cite where this methodology has been validated? Why would it be even useful to search the grey literature when it would likely fall below the cutoff with a lack of information? Dear, 

Regarding the methodological validity, a plethora of information and published articles utilizing related methodology of systematic review and meta-analysis can be accessed from reputable journals and publishers worldwide. 

Besides, to keep the scientific integrity of this systematic review and meta-analysis, we applied the following tools and protocols which are duly cited in the method section. 

This method (protocol) has been registered on PROSPERO, University of York with ID: CRD42020175168 and available online at:

https://www.crd.york.ac.uk/PROSPERO/display_record.php?ID= CRD42020175168&ID= CRD42020175168.

PRISMA flow diagram

PRISMA checklist

Joanna Briggs institute, University of Adelaide, Australia, risk of bias (quality) assessment tools 

Publication bias assessment tools (e.g Begg’s and Mazumder) 

In systematic review and meta-analysis, searching for grey literature (unpublished thesis/dissertation, online repositories and related documents) to address relevant data for the research question IS MANDATORY. If we had considered only published articles or articles in legitimate database only, serious publication bias (funnel plot asymmetry) would have occurred for all tests. JBI and Cochrane recommend searching grey literature to minimize the publication bias.

Studies should be excluded from analysis by their quality and relevance not by their publication status. 

Any study that did not address the primary outcome of interest (EPSE and/or its components) could not be eligible for further process. 

7. Did you consider sub-analyses by medication type? Diagnosis? Age? Dear,

We have initially considered subgroup analysis on both demographic and clinical factors. Nevertheless, the high heterogeneity of diagnosis (Schizophrenic, acute psychotic, SMI, psychotic with SUD, etc…) and medication use (typical, atypical and mixed use) hindered us from doing meta-analysis. Instead, we have included the age of the patients and drug regimen issues in the table 1 describing the study characteristics

Fortunately, all the patients included in the study are adults and adolescents and no clear cut-off point to run subgroup analysis.

8. a table of characteristics of your final included studies is necessary. 

Dear,

We have included Table 1 in the manuscript next to the paragraph that narrate what the characteristics of the studies are all about. 

We have also amended the study characteristics section in much more elaborated way. 

9. why 11 studies in the main meta-analysis when 15 included? More clarity needed. Dear reviewer, 

As we put the inclusion and exclusion criteria, original studies that address the overall EPSE only and/or one of its components are eligible for systematic review. Nevertheless, only studies with homogenous outcome measures should be retained for further quantitative synthesis (meta-analysis) accordingly.

In this regard, few studies did not address the entire EPSE rather focused on specific symptoms such as TD or DIP. 

In line with, the authors have clarified and made critical revision on the write up and outcome measures of included studies. 

10. where are the I2 heterogeneity statistics throughout the results? � The measure of heterogeneity (I2 statistics) is available in all forest plots (Figure 2 to Figure 6). We tend to focus on major finding in this study rather than narrating everything on the figures and table. 

11. more consideration and detail is needed regarding the included study antipsychotic type given their importance as detailed in the introduction. Dear,

Please check the revised manuscript in this regard.

We have undergone an extensive edition throughout the document with much emphasis to introduction and result sections. In line with this, we included a column describing the type of antipsychotic medications taken and respective regimens in the table describing study characteristics. 

Reviewer 2

Title

1. Does the title give clear idea about the article? Yes Thank you for your constructive comments 

Abstract

2. Does the abstract concisely describe the content and scope of the project and identifies the project’s objective, its methodology and its findings, conclusions, or intended results? No

Under background, add something about mood disorder with psychotic. Because they also take antipsychotic medicine.

Under conclusion, you tried to conclude as appropriate prevention of EPSE is important. How you could come across with this recommendation? Dear reviewer, 

Under the background section of abstract, mood disorder with psychotic feature was also included. Clinically heterogenous group of patients (with schizophrenic, schizoaffective, acute psychotic with substance use disorders, bipolar disorders, those with severe mental illnesses e.tc) were considered as far as they took antipsychotic medication. The objective of this study is to estimate antipsychotic-induced EPSEs regardless of neuropsychopathological features. 

We have tried to rephrase the conclusion and limitation sections of the manuscript as well. 

Please clarify the statement “They are classified as first generation or conventional (typical) antipsychotics (FGA) and second generation (atypical) antipsychotics (SGA)” We have amended it along with other introduction sections 

Introduction

3. Does the introduction give clear idea about the article? YES

Use similar word for antipsychotic/s through the document. Also for EPS/EPSE. Use similar word and abbreviation. Sometimes you say EPS, and sometimes EPSE/s

What do you mean EPSs? The end of paragraph of introduction

What do you say by side effects which appear when anticholinergic medicine removed from the patient? Dear,

As per your request, we have used abbreviations consistently throughout the document. We preferred to use EPSE to EPS

For terms which were repeated for at least three times, the full term with abbreviation in the parenthesis was used at the first appearance and the abbreviation only thereafter. 

Kindly check the amendments (track-changes) made on the introduction section to clarify the rationale of this study. We have included the role of anticholinergics on the antipsychotic drug therapy. 

Methodology

4. Did methodology part is clear? No

Well, if specified who (author/s) were extracted data.

What do you say about data quality assessment?

Which model did you use to determine pooled prevalence?

Is all original paper report standard error? If no, what is your action? If yes, please specify.

What is about independent factors? 

Dear reviewer, 

We have made extensive revision on the methodology as well. 

We included the initials of authors who were involved in data source searching, study selection process, data extraction and risk of bias (quality) assessments. 

Considering the high heterogeneity of study (primarily variation in study characteristics) triggered us to utilize random effects pooling model with inverse variance method. This is clearly specified in the data synthesis and analysis 

In practical sense, standard error is not mandatory to estimate the pooled outcome measures of this data type. Sample size and event rate are the two key variables to be used. However, to determine publication bias, meta-bias, meta regression etc, event rate and standard error of event rate are two important data sets. To do so, STATA can automatically generate the SE (seES) during computation or else we can covert it to standard error manually if confidence interval is given (in the absence of sample size).

o (UCI-LCI)/(2*invnormal)(0.975))

The secondary outcome measures (related to independent factors) were not considered in this study due to inconsistency of findings of included studies. It is illogical to meta-analyze several explanatory variables when the method of analysis of individual studies is diverse. 

Results

5. Are results clear and appropriate with title? Yes

6. Revise the references as per the journal guideline

7. Generally, the paper is interesting. The most problems were found around methodology part. It needs major revision.

8. The paper needs an English language copy editing from the beginning to the end. Please focus on it. Dear, 

The references are revised as per the journal guideline 

Generally, questions, comments and suggestions that were focusing in the methodology part are addressed accordingly. We have also undergone serious language editing (syntax, normalization, logical follow, cohenrence and consistency) over the last one month as well. 

Reviewer 3

Reviewer #3: I greatly appreciate the reviewing process and would like to propose designing a further review on the antipsychotic-induced dystonia and the relative contribution of each antipsychotic to the side effect specified in addition to the dose-related side effect threshold Thank you. 

We will try to design a further review focusing on the antipsychotic-induced dystonia and the relative contribution of each antipsychotic agents to the side effect specified to generate evidence-based medicine in the area of antipsychotic therapy. 

Regards, 

Authors

---

## [Decision Letter · Decision Letter 1]

30 Apr 2021

PONE-D-20-32247R1

Antipsychotic-induced extrapyramidal side effects: A systematic review and meta-analysis of observational studies

PLOS ONE

Dear Dr. Ali,

Thank you for submitting your manuscript to PLOS ONE. After careful consideration, we feel that it has merit but does not fully meet PLOS ONE’s publication criteria as it currently stands. Therefore, we invite you to submit a revised version of the manuscript that addresses the points raised during the review process.

We look forward to receiving your revised manuscript.

Kind regards,

Ahmed Negida, MD

Academic Editor

PLOS ONE

Journal Requirements:

Reviewers' comments:

Reviewer's Responses to Questions

**Comments to the Author**

1. If the authors have adequately addressed your comments raised in a previous round of review and you feel that this manuscript is now acceptable for publication, you may indicate that here to bypass the “Comments to the Author” section, enter your conflict of interest statement in the “Confidential to Editor” section, and submit your "Accept" recommendation.

Reviewer #1: All comments have been addressed

Reviewer #2: (No Response)

Reviewer #3: All comments have been addressed

2. Is the manuscript technically sound, and do the data support the conclusions?

Reviewer #1: Yes

Reviewer #2: Partly

Reviewer #3: Yes

3. Has the statistical analysis been performed appropriately and rigorously? 

Reviewer #1: Yes

Reviewer #2: N/A

Reviewer #3: Yes

4. Have the authors made all data underlying the findings in their manuscript fully available?

Reviewer #1: Yes

Reviewer #2: Yes

Reviewer #3: Yes

5. Is the manuscript presented in an intelligible fashion and written in standard English?

Reviewer #1: Yes

Reviewer #2: No

Reviewer #3: Yes

6. Review Comments to the Author

Reviewer #1: The authors have done a good job revising their manuscripts and improving the clarity of methods and results. I have no further comments

Reviewer #2: Comments

1. In the previous evaluations or comments the author(s) reacts for raised comments superficially. Or the author(s) was (were) not answer for reviewers’ comments. For example, EPSEs which is under introduction page 3 paragraph 1 line 5, and EPS which is on the same page last paragraph line 3 were not similar. Well if the author(s) incorporate deeply for the raised questions.

2. Under results (search findings), the end line of first paragraph page 7, you were exclude 43 papers because various reason. Could explain these various reasons?

3. The publication years of the included studies ranged from 2000 to 2019. I think it is too old the papers you were used from 2000.

4. Papers included in the study form both America and Africa is too few. This implies that as not enough studies were conducted with same title. Could you explain?

5. Title of the table needs modification. Table 1 page 9

6. An article you were used in the raw 10 column 2 is not clear. Which said South Africa and Nigeria

7. Please modify the strength and limitation of the study separately. It is not clear as it is.

Reviewer #3: Thank you for this informative review and would like you to consider the future focused review as mentioned before.

7. PLOS authors have the option to publish the peer review history of their article (what does this mean?). If published, this will include your full peer review and any attached files.

Reviewer #1: No

Reviewer #2: No

Reviewer #3: **Yes: **MM

---

## [Author Response · Author response to Decision Letter 1]

2 Jun 2021

All the comments and questions are incorporate in the revised manuscript

---

## [Decision Letter · Decision Letter 2]

25 Aug 2021

Antipsychotic-induced extrapyramidal side effects: A systematic review and meta-analysis of observational studies

PONE-D-20-32247R2

Dear Dr. Ali,

We’re pleased to inform you that your manuscript has been judged scientifically suitable for publication and will be formally accepted for publication once it meets all outstanding technical requirements.

Kind regards,

Xenia Gonda

Academic Editor

PLOS ONE

Additional Editor Comments (optional):

Reviewers' comments:

Reviewer's Responses to Questions

**Comments to the Author**

1. If the authors have adequately addressed your comments raised in a previous round of review and you feel that this manuscript is now acceptable for publication, you may indicate that here to bypass the “Comments to the Author” section, enter your conflict of interest statement in the “Confidential to Editor” section, and submit your "Accept" recommendation.

Reviewer #1: All comments have been addressed

Reviewer #3: All comments have been addressed

2. Is the manuscript technically sound, and do the data support the conclusions?

Reviewer #1: Yes

Reviewer #3: Yes

3. Has the statistical analysis been performed appropriately and rigorously? 

Reviewer #1: Yes

Reviewer #3: Yes

4. Have the authors made all data underlying the findings in their manuscript fully available?

Reviewer #1: Yes

Reviewer #3: Yes

5. Is the manuscript presented in an intelligible fashion and written in standard English?

Reviewer #1: Yes

Reviewer #3: Yes

6. Review Comments to the Author

Reviewer #1: The authors have made revisions according to the comments from the previous round of review. I believe these revisions have made the manuscript more clear and answered the issues that were raised

Reviewer #3: I appreciate the manuscript and encourage the authors to extend the study as previously recommended in a further review.

7. PLOS authors have the option to publish the peer review history of their article (what does this mean?). If published, this will include your full peer review and any attached files.

Reviewer #1: No

Reviewer #3: **Yes: **MM

---

## [Editor Report · Acceptance letter]

1 Sep 2021

PONE-D-20-32247R2 

Antipsychotic-induced extrapyramidal side effects: A systematic review and meta-analysis of observational studies 

Dear Dr. Ali:

I'm pleased to inform you that your manuscript has been deemed suitable for publication in PLOS ONE. Congratulations! Your manuscript is now with our production department. 

Kind regards, 

on behalf of

Dr. Xenia Gonda 

Academic Editor

PLOS ONE